# Modeling the Influence of Engine Dynamics on Its Indicator Diagram

**DOI:** 10.3390/s21237885

**Published:** 2021-11-26

**Authors:** Piotr Deuszkiewicz, Jacek Dziurdź, Paweł Fabiś

**Affiliations:** 1Institute of Machine Design Fundamentals, Warsaw University of Technology, 02-524 Warszawa, Poland; jacek.dziurdz@pw.edu.pl; 2Department of Transport and Aviation Engineering, Silesian University of Technology, 40-019 Katowice, Poland; pawel.fabis@polsl.pl

**Keywords:** internal combustion engine, indicator diagram, dynamics

## Abstract

This article presents a proposal to describe the pressure changes in the combustion chamber of an engine as a function of the angle of rotation of the crankshaft, taking into account changes in rotational speed resulting from acceleration. The aim of the proposed model is to determine variable piston forces in simulation studies of torsional vibrations of a crankshaft with a vibration damper during the acceleration process. Its essence is the use of a Fourier series as a continuous function to describe pressure changes in one cycle of work. Such a solution is required due to the variable integration step during the simulation. It was proposed to determine the series coefficients on the basis of a Fourier transform of the averaged waveform of a discreet open indicator diagram, calculated for the registration of successive cycles. Recording of the indicative pressure waveforms and shaft angle sensor signals was carried out during tests on the chassis dynamometer. An analysis of the influence of the adopted number of series coefficients on the representation of signal energy was carried out. The model can also take into account the phenomenon of work cycle uniqueness by introducing random changes in the coefficients with magnitudes set on the basis of determined standard deviations for each coefficient of the series. An indispensable supplement to the model is a description of changes in the engine rotational speed, used as a control signal for the PID controller in the simulation of the load performed by the dynamometer. The accuracy of determining the instantaneous rotational speed was analyzed on the basis of signals from the crankshaft position angle sensor and the piston top dead center (TDC) sensor. Limitations resulting from the parameters of digital signal recording were defined.

## 1. Introduction

One of the problems of the operation of a combustion engine is high amplitudes of torsional vibrations that can damage the shaft [1,2]. This problem is particularly critical for engines with a large number of cylinders where there are many forms of natural vibrations. One of the methods of limiting the magnitude of torsional angles is the use of torsional vibration dampers [3,4,5,6]. The magnitude of torsional vibrations results from excitations caused by gas-dynamic and inertial forces related to engine operation. Damping of torsional vibrations is particularly important in the case of excitations close to the range of resonant vibrations, which increases engine reliability and durability. The main problem in the operation of a damper is to make a selection of parameters to avoid, or minimize the time of, operation in resonance conditions [7,8,9,10]. The effectiveness of damper operation is verified by measuring torsional vibrations of the shaft during engine acceleration.

The authors conducted a study of the VR6 3.6 FSI gasoline engine with 280 HP at 6250 rpm and torque of 370 Nm at 3500 rpm, installed in the Volkswagen Passat. A chassis dynamometer operating in acceleration mode was used to determine the power and torque characteristics of the engine as a function of rotational speed. Measurements of torsional vibrations of the crankshaft journal with the damper attached were performed with a Polytec RLV-5500 laser vibrometer [3]. On the basis of the dimensions of the piston–crank system elements, a model was developed, which was then used to determine the system of ordinary differential equations. These equations were introduced into MATLAB Simulink. It was established that for the correct operation of the model, it is necessary to describe the changes in the values of the forces that force motion (forces acting on the pistons) and to describe the changes in the rotational speed of the engine, which will be used to control the load by the PID controller (simulation of the load implemented by the chassis dynamometer).

Therefore, it was decided to develop a method of modelling pressure changes in the combustion chamber of an engine as a function of rotational speed and the angle of the crankshaft position, developed on the basis of measurements carried out during acceleration on an engine test stand or chassis dynamometer. On this basis, a force acting on the surface of a piston, proportional to the pressure and changing as a function of the angle of rotation, can be determined. This force can be converted into transverse and torsional loads of the modelled crankshaft [11,12]. This model can be used as a dynamic excitation in simulation tests of torsional vibrations of crankshafts with a torsional vibration damper. The application of an excitation similar in nature to the real one will make it possible to fine-tune the damper parameters on the basis of a computer simulation [13,14].

The analysis of cylinder pressure changes has long been an effective tool in the research and development of combustion engines. The development of measurement techniques made it possible to increase the accuracy of measurements and facilitated their analysis. Closed and open indicator diagrams created on the basis of measurements contain information related to all phases of operation: suction and compression, ignition and expansion, and exhaust. Contrary to ideal theoretical diagrams, they take into account the randomness of the combustion process, although they can be used to verify the developed theoretical characteristics using averaging.

The analysis of indicator diagrams is described comprehensively in the literature. On their basis, the influence of fuel used [15,16,17], ignition timing [18], valve opening and closing [19], or the technical condition [12,20,21] on the combustion process are determined. On the basis of pressure in the combustion chamber, the loads on the mechanical components of an engine are determined. Some of these studies concern tests under constant speed conditions, but there are also studies related to transient conditions [22,23]. The modeling proposal presented in the article may also be useful for this type of analysis.

## 2. Description of the Conducted Research

The measurements were carried out on a BOSCH FLA 203 2WD chassis dynamometer with an eddy current brake. Basic parameters of the dynamometer were:maximum power (Static)—260 kW (353 HP) at 260 km/h,maximum power (Dynamic)—300 kW (408 HP),maximum speed—260 km/h.

For the research, a 4-cylinder, 4-stroke gasoline engine with a capacity of 1600 cm^3^ of the Opel Astra car, with a power of 55 kW at 5200 rpm and torque of 128 Nm at 2800 rpm was used. The reason for choosing this engine was its preparation for research, consisting in the installation of appropriate measuring systems.

The measuring system for continuous signal recording consisted of:A KISTLER 6121 piezoelectric pressure sensor with a KISTLER 5011 preamplifier, screwed into the hole in the engine head above one of the cylinders;A crankshaft position sensor generating pulses every 1 arc degree and a KISTLER 2613B top dead center (TDC) sensor;A system for recording signals, enabling synchronous discrete recording of data: Brüel&Kjaer LAN-XI with modules 3050-B-6/0.

As part of the research, measurements were made for the following engine operating parameters:Operation at a “constant” rotational speed in gears II, III, and IV: 1500, 2000, 2500, 3000 and 3500 rpm;Acceleration in gear IV for different positions of the accelerator pedal: 50, 70, and 100% from the rotational speed of approximately 1200 rpm to approximately 5900 rpm.

The value of the load, realized by the chassis dynamometer, was determined by the dynamometer control system on the basis of data from the vehicle’s OBD system: acceleration pedal settings and engine speed.

## 3. Analysis of Signals from Crankshaft Position Sensors

Indicator diagrams of the combustion engine show the pressure change as a function of the angle of the crankshaft position. The position is determined on the basis of a signal recorded in time by a sensor [24,25] changing the voltage value from 0 to 5 V with successive pulses (TTL signal). The accuracy of determining the occurrence of the pulses over time, limited by the sampling frequency, which in the presented examples was *f_s_* = 131,072 Hz (the maximum possible frequency in the measuring system used) is of significant importance. The absolute error Δ*t* in determining the point in time is equal to half the time resolution d*t* being the opposite of the sampling frequency *f_s_*:$$\Delta t\;=\;\frac{1}{2}dt\;=\;\frac{1}{2f_{s}}\;=\;3.815\;\mu s$$

This is a relatively small value, but its influence on the accuracy depends on the rotational speed of an engine and the number of pulses per revolution. The accuracy of representing the pressure waveform as a function of the angle of rotation can be determined on the basis of comparison of successive cycles of work under set conditions (constant rotational speed and constant load). As the combustion process is a random phenomenon, the comparison should be limited to the part of the indicator diagram related to the suction and compression strokes, which should be repeatable, assuming that the error in determining the value of pressure in this case is negligible. Figure 1 shows example registrations of 100 consecutive cycles (200 shaft revolutions) for the rotational speed *n_rev_* = 3500 rpm.

In order to assess the accuracy, the standard deviation of the part related to the suction and compression stroke (shaft rotation angle from 0° to 360°) was calculated. For the sake of comparison, the standard error was also calculated for the rotational speed of 1500 rpm and 2000 rpm. The results are presented in Figure 2.

The standard deviation, which is a measure of uniqueness, is relatively small in terms of the angles of rotation until ignition occurs. It can be assumed that although sampling affects the accuracy of the representation of the diagram, it is not so important in the considered cases. However, it should be remembered that this error will grow along with an increase in rotational speed.

Another important issue is the determination of the instantaneous value of rotational speed *n_rev_* (Equation (1)), which is a necessary parameter of the proposed model. It can be determined on the basis of time between successive pulses of signals from sensors. For the signal related to the top dead center of the piston (1 pulse per revolution), this time corresponds to the angle *φ*_1_ = 2 π rad, and for the crankshaft position signal (360 pulses per revolution), this time corresponds to the angle *φ*_2_ = 17.45 mrad.
(1)$$n_{rev\_i}\;=\;\frac{60}{2\pi }\frac{\varphi_{1}}{t_{i}\;-\;t_{i-1}}\;\mathrm{or}\;n_{rev\_j}\;=\;\frac{60}{2\pi }\frac{\varphi_{2}}{t_{j}\;-\;t_{j-1}}\frac{1}{360}$$
where:
*t_i_*–*t_i_*_−1_        time between successive impulses from the TDC sensor,*t_j_*–*t_j_*_−1_        time between successive impulses from the sensor of the crankshaft position.

The absolute error of determining the times *t_i_*–*t_i_*_−1_ and *t_j_*–*t_j_*_−1_, resulting from sampling, is equal to time resolution d*t* = 7.629 μs. The values of the relative errors in determining the times between successive pulses for different rotational speeds of the engine are presented in Table 1 and Table 2.

The presented errors in determining the time between successive pulses affect the accuracy of determining the instantaneous rotational speed. Due to excessive relative errors, the signal from the shaft position sensor is not useful for determining the instantaneous rotational speed.

Because the proposed model will be based on the determination of pressure in the work cycle, the instantaneous rotational speed for this cycle can be determined, with sufficient accuracy, on the basis of the signal from the sensor of the top dead center of the piston. Figure 3 shows example diagrams of speed changes with acceleration determined for both sensors.

By treating the error caused by sampling as an effect of random disturbance, its impact can be reduced by filtering the waveform of the determined rotational speed with an appropriately adjusted low-pass filter (Figure 3). This method is especially useful for keeping rotational speed constant, but can also cause a sufficiently good effect at variable speed. The effectiveness of this method was confirmed in tests, where on the basis of such a waveform obtained, the signals were resampled [26].

## 4. Modeling the Pressure Signal in the Cylinder of a Combustion Engine on the Basis of Measurements

Knowledge of the instantaneous pressure value in an engine combustion chamber allows determination of a force acting on the piston as a function of the angle of the crankshaft revolution. On this basis, forces and moments in the crank system can be determined.

Simulation tests of physical system models, carried out, e.g., in programs such as MATLAB Simulink, require a variable integration step [27]. This is selected during the analysis and its magnitude depends on the state of the solution. Pressure changes as a function of shaft revolution, obtained from measurements, are discrete values for the crankshaft rotation angles determined by the measuring system. Therefore, they cannot be directly used in the simulation but, on their basis, a model can be developed that meets the following assumptions:The magnitude of the modelled pressure should be at least a function of rotational speed and the angle of rotation of the crankshaft for the selected gearbox setting;The pressure change should be described by a relatively simple function that allows the values to be determined for any angle of rotation of the crankshaft;The pressures at the end of the cycle and at the beginning of the next cycle should be similar (with the smallest possible difference).

These assumptions are met by the approximation of discrete pressure values using a Fourier series determined for the full cycle of operation of a four-stroke engine (two revolutions—720°). The series coefficients can be determined using relationships between a Fourier series and a Fourier transform [28], assuming that a single execution of the cycle presents the waveform of the periodic function for the basic period equal to 720° (two shaft revolutions). These coefficients can be determined with sufficient accuracy using the spectrum of a discrete cycle.

The explanation of the model creation process is presented using the example of the analysis of measurements carried out with a “constant” rotational speed of *n_rev_* = 3500 rpm for 100 consecutive work cycles (200 crankshaft revolutions). Figure 4 shows the waveforms of the analyzed cycles along with the average cycle obtained by averaging in the time domain.

The discrete notation of pressure consists of *K* = 720 points. The two-sided complex spectrum, calculated using a Fourier transform (FFT), will have 360 points (*K*/2) for the positive part. This is the maximum number of pairs of coefficients (*a_n_*, *b_n_*) and coefficient *a*_0_, which can be used in the discrete form of a Fourier series described by the equation:(2)$$x_{k}\;=\;a_{0}\;+\;\sum_{n\;=\;1}^{N}\left(a_{n}\cos\left(2\pi n\frac{k}{K\;-\;1}\right)\;+\;b_{n}\sin\left(2\pi n\frac{k}{K\;-\;1}\right)\right).$$

Because computation time is a significant problem in computer simulations, one should consider reducing the number of points from which Fourier series coefficients will be determined. One of the criteria that should be met in this case is the condition of maintaining the energy of the process with sufficient accuracy. In the presented example, the analysis was carried out on the basis of the discrete power spectrum (equivalent to the energy of the process considered in time) for the averaged cycle, assuming that the energy represented by the selected number of spectral components should not differ from the total energy by more than 1%. Figure 5 shows the power spectrum of the averaged cycle with marked points corresponding to the numbers of points *N* = 10 and *N* = 20. The number of points was selected on the basis of the analysis of the difference between the signal power for all spectral points and the power of the signal limited to *N* points (Figure 6) and comparing the functions described by a Fourier series with the averaged cycle. The first 10 points of the spectrum were sufficient to meet the energy condition (decrease in relation to the total signal power 0.78%). However, after comparing with the average cycle (Figure 7), the first 20 points of the spectrum were assumed.

In order to correctly carry out the process of determining Fourier series coefficients on the basis of the discrete complex spectrum, one should take into account the operation of algorithms used to calculate the discrete form of a Fourier transform (DFT), and in practice, to determine a fast Fourier transform (FFT). In computational algorithms, including the one used in MATLAB, we obtain *K* points of the two-sided complex spectrum from *K* points of a discrete waveform [28]. One should note, however, that the points of the positive part of the spectrum (formally corresponding to the positive frequencies in the continuous spectrum) are shown at the beginning and they are followed by points of the negative part. For the sake of clarity of the description, Figure 8 shows a complex spectrum module, in which the value *k* denotes the number of the spectral point.

The computed discrete complex spectrum is characterized by the following properties:The point with index *k* = 0 (corresponds to the frequency equal to 0 Hz) represents the average value of the analyzed signal (occurs once in the spectrum);The point with index *k* = *K*/2 corresponds to the value of the component with the Nyquist frequency *f_N_* (occurs once in the spectrum);The points of the positive part for $k\;=\;\left(1,\;2,\;3,\;\ldots \;\frac{K}{2}\;-\;1\right)$ have their counterparts in the negative part for $k\;=\;\left(K\;-\;1,\;K\;-\;2,\;K\;-\;3,\;\ldots \;\frac{K}{2}\;+\;1\right)$. The points corresponding to each other have the same amplitude value, equal to half the amplitude of the harmonic component of frequency k·Δf, but the phase angle of the opposite sign—they are complex conjugate numbers.

Assuming the notation of a discrete Fourier transform in the form:(3)$$X_{k}\;=\;\frac{1}{K}\overset{K\;-\;1}{\underset{j\;=\;1}{\sum }}x_{j}e^{\;-\;i2\pi j\frac{k}{K}}$$
one can record the interdependencies connecting *N* coefficients of a Fourier series with the points of a discrete Fourier transform:(4)a0=X0a1…N−1=2·ReX1…N−1b1…N−1=−2·ImX1…N−1

The sign “–” in the notation of the imaginary part takes into account that, in the positive part of the two-sided complex spectrum, the values of the imaginary part are equal to $-\frac{b_{n}}{2}$. An example of a complex spectrum of harmonic functions is shown in Figure 9.

In the presented example, *N* = 20 Fourier series coefficients were determined for each of the 100 consecutive work cycles, and then they were averaged. In this way, the coefficients for the average cycle were determined (Figure 10). The function describing the averaged cycle was calculated for the points corresponding to the successive shaft positions in accordance with the measurements, and the obtained waveform was compared with the averaged waveform in the time domain (Figure 7). The values of the coefficients *a_n_* and *b_n_* and the standard deviations *σa_n_* and *σb_n_* are presented in Appendix A in Table A1.

The combustion process in the cylinder is a dynamic phenomenon that causes the uniqueness of work cycles [29,30,31], which can be seen in Figure 4. This is due, among other things, to an uneven filling of the cylinder during the intake stroke and an uneven course of the combustion process during the power stroke. Additionally, it can be influenced by the control system which changes the ignition advance angle and the dose of the injected fuel. Taking these differences into account, the model should be supplemented with a random change in Fourier series coefficients *a_n_* and *b_n_* within the range resulting from the determined values of standard deviations *σ**a_n_* and *σb_n_* (Figure 10).

In order to accelerate the operation of a pressure calculation algorithm, another notation form of a Fourier series can be used:(5)$$x_{k}\;=\;a_{0}\;+\;\sum_{n\;=\;1}^{N}\left(A_{n}\cos\left(2\pi n\frac{k}{K\;-\;1}\;+\;\Phi_{n}\right)\right).$$
where:

$A_{n}\;=\;\sqrt{a_{n}^{2}\;+\;b_{n}^{2}}$.

$\Phi_{n}\;=\;-\mathrm{atan}\left(\frac{b_{n}}{a_{n}}\right)$.

The sign “–” when determining the phase shift angle *Φ_n_* results from the same reason as when determining coefficients *b_n_*. The time of calculations performed on the basis of Formula (5) is more than two times shorter than the calculations carried out on the basis of Formula (4) because the determination of the value of the trigonometric function occurs only once. Figure 11 shows *A_n_* and *Φ_n_* coefficients of the series and the values of standard deviations *σA_n_* and *σ**Φ_n_*.

The values of coefficients *A_n_* and *Φ_n_* and standard deviations *σA_n_* and *σ**Φ_n_* are presented in Appendix A in Table A2.

The determined discrete waveform of the average cylinder pressure can also be used to determine the engine power of *N_it_*. Values of the force acting on the piston as a function of the angle of rotation of the shaft *φ* (in discrete form for subsequent positions: $\varphi_{k}\;=\;k\;\cdot \;1\;\mathrm{arc}\;\mathrm{degree}$) are:(6)$$F_{k}\;=\;p_{k}\;\cdot \;A_{p}$$
where:

*A_p_*—the surface area of the piston crown.

The displacement of the piston as a function of the angle of rotation of the shaft depends only on the geometrical sizes of the piston–crank system. Figure 12 shows the displacement of the piston *x_p_* depending on the angle of rotation for the engine under test. The displacement of the piston for subsequent values of the angle *φ_k_* is:(7)$$\Delta_{k}\;=\;x_{p\_k\;+\;1}\;-\;x_{p\_k}$$
and indicated work *W_i_k_*:(8)$$W_{i\_k}\;=\;F_{k}\;\cdot \;\Delta_{k}$$

The indicated work *W_i_* for one cylinder is the sum of the values of the *W_i_k_*. Figure 12 shows the *W_i_k_* values for subsequent angles of rotation of the shaft.

Taking into account the cycle time *t_i_*_+1_–*t_i_*, the number of cylinders *n* = 4, and the number of revolutions per duty cycle *z* = 2, the average engine power can be determined (assuming that the cylinder operating parameters are similar):(9)$$N_{it}\;=\;n\frac{W_{i}}{t_{i\;+\;1}\;-\;t_{i}}$$
where:

$t_{i\;+\;1}\;-\;t_{i}\;=\;\frac{60}{n_{rev}}z$—is formally determined from the TDC sensor signal.

In the example shown, the value of the indicated power *N_it_* = 45.8 kW is obtained. For comparison, the value of power on the wheels, recorded by the measuring system of the chassis dynamometer, amounted to approx. 36 kW. The difference is due to the resistances occurring in the drive system, the rolling resistance of the drive wheels, and errors in the estimation of the indicated power.

## 5. Modelling of Pressure Changes during Acceleration

The construction of a pressure change model carried out for engine acceleration is a more difficult issue. It is not possible to use a large number of averages, like for constant rotational speed, because speed changes with time, and during the cycle. Based on the graph of speed changes over time, it was found that the maximum increase in rotational speed per one work cycle is approx. 20 rpm (for the rotational speed of approx. 1500 rpm). Therefore, in the presented example, 10 consecutive cycles were assumed for averaging—five before the adopted rotational speed and five after. The choice of rotational speeds for the description of pressure changes is limited by the duration of subsequent cycles. In the presented example, for the sake of clarity of the presentation, speeds in the range from 1300 to 5800 rpm were selected, changing every 100 rpm. The waveform of changes in rotational speed together with the values selected for the analysis are shown in Figure 13.

Figure 14 shows the first 20 coefficients *a_n_* and *b_n_* of a Fourier series determined for selected rotational speeds. The practical use of the presented waveforms in the proposed model to determine the values of the coefficients for any value of rotational speed can be carried out in two ways:Approximate each graph of coefficients *a_n_* and *b_n_* by a polynomial;Two points with the closest values can be used to determine the values of coefficients *a_n_* and *b_n_* for the required rotational speed.

Figure 15 shows a comparison of the averaged waveforms of pressure changes determined by a Fourier series for selected rotational speeds, and in Figure 16 the same waveforms showing pressure changes as a function of the shaft rotation angle *φ* and rotational speed of the shaft *n_rev_*.

Using Formulas (6)–(9), the values of the power estimated *N_it_* as a function of engine speed *n_rev_* were determined. Figure 17 shows a comparison of the power values on the wheels with the determined values. The obtained results are influenced by accuracy of determining rotational speeds from the TDC sensor signal, which is particularly visible for higher speed values.

## 6. Conclusions

The proposed method of describing pressure changes in the combustion chamber of an engine will have practical application. Its development had to take into account the required simplicity of modeling, which can be used in computer simulation, while maintaining sufficient accuracy of mapping. The authors propose to use it in modeling the forces forcing the movement of the piston–crank system in simulation studies. In order to be able to compare the results of torsional vibration measurements of the real object with the results obtained from the model, it is necessary to model the load on the system carried out by the chassis dynamometer. For this reason, it is necessary to describe the changes of engine rotational speed as a function of time, which will be used as a signal to control the PID regulator in the load model. In both cases, it is important to accurately reproduce the changes in values as a function of the crankshaft rotation angle or time.

Limiting the description of the pressure changes estimated to the first 10 coefficients of the Fourier series causes a change in energy by about 1%, but in order to more accurately reproduce the shape of the waveform, it is proposed to use the first 20 coefficients. This will extend the duration of the simulation, but will avoid additional time-varying forces resulting from inaccuracies in the description based on the first 10 coefficients.

An important problem is the impact of the accuracy of determining the instantaneous engine rotational speed based on the registration of discrete signals from the shaft position sensor (360 pulses per revolution) and the TDC sensor (1 pulse per revolution). Of great importance in this case is the sampling rate limited by the capabilities of the measuring equipment used. Based on the results of the considerations (Table 1 and Table 2), it can be clearly stated that a large number of pulses per revolution does not improve the accuracy of mapping the time necessary to determine the rotational speed. Therefore, it was found that only the signal from the TDC sensor could be used to determine the changes in rotational speed due to the small relative error.

It should also be noted that the error in determining the angle of the crankshaft position may affect the accuracy of the indication pressure waveform, especially for higher engine rotational speed values.

The research used a set of measuring sensors that are also used for other tests related to the operation of internal combustion engines. Therefore, the proposed method is not limited only to spark-ignition gasoline engines, but also to compression-ignition, CNG, or LNG engines. The authors believe that the proposed method can be used to study the impact of the fuel used on the dynamics of the car’s acceleration, or for statistical analysis related to research on the impact of the fuel used, or control parameters on the operation of the engine or diagnostics of the combustion process, especially in undetermined operation conditions.

## Figures and Tables

**Figure 1 sensors-21-07885-f001:**
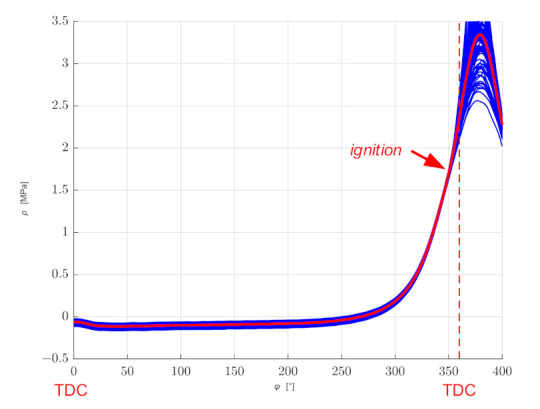
Example waveforms of 100 consecutive engine work cycles and an average waveform for constant operating conditions.

**Figure 2 sensors-21-07885-f002:**
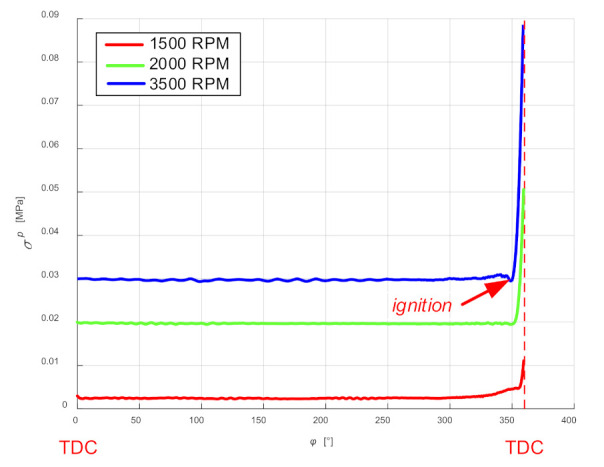
Comparison of the values of standard deviations for the waveforms from Figure 1 and for measurements with the rotational speed of 1500 and 2000 rpm, determined in the range of the angle of rotation from 0° to 360°.

**Figure 3 sensors-21-07885-f003:**
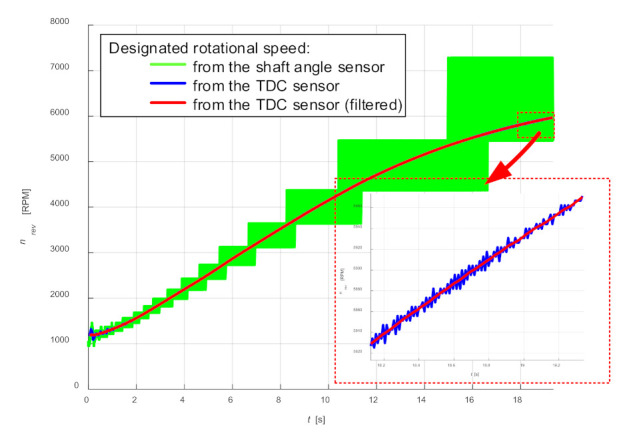
Example diagrams of rotational speed changes determined on the basis of signals from the sensor of the top dead center of the piston and the sensor of the crankshaft position.

**Figure 4 sensors-21-07885-f004:**
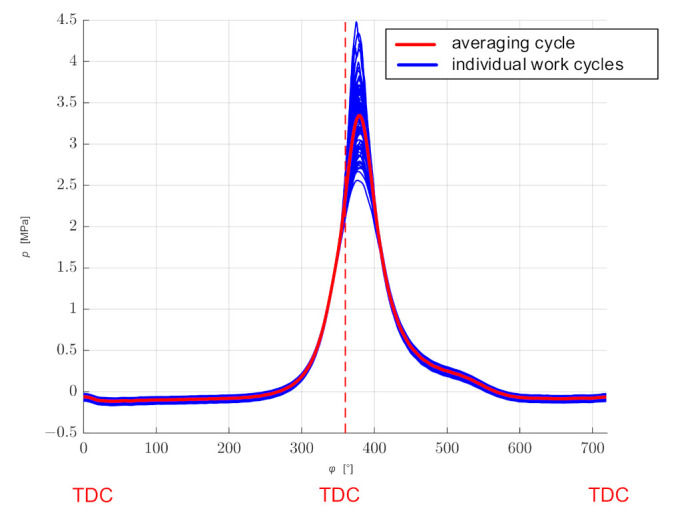
The waveform of 100 consecutive work cycles and the average cycle.

**Figure 5 sensors-21-07885-f005:**
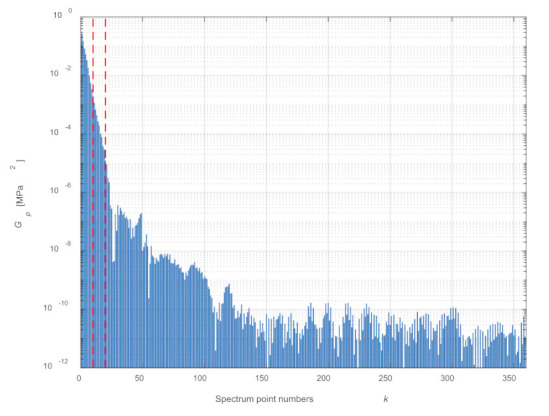
Averaged cycle power spectrum.

**Figure 6 sensors-21-07885-f006:**
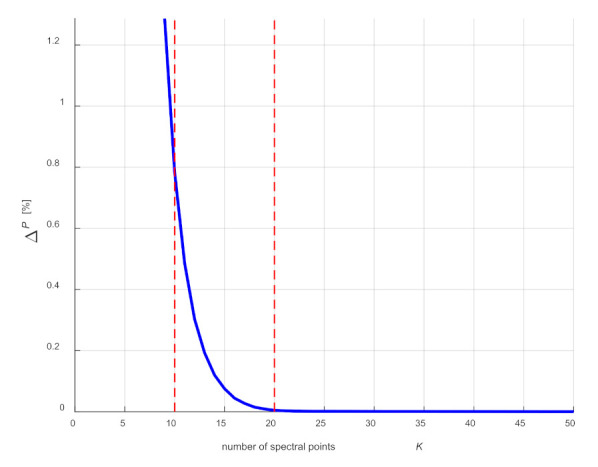
Change in the signal energy depending on number *K* of the assumed spectral points.

**Figure 7 sensors-21-07885-f007:**
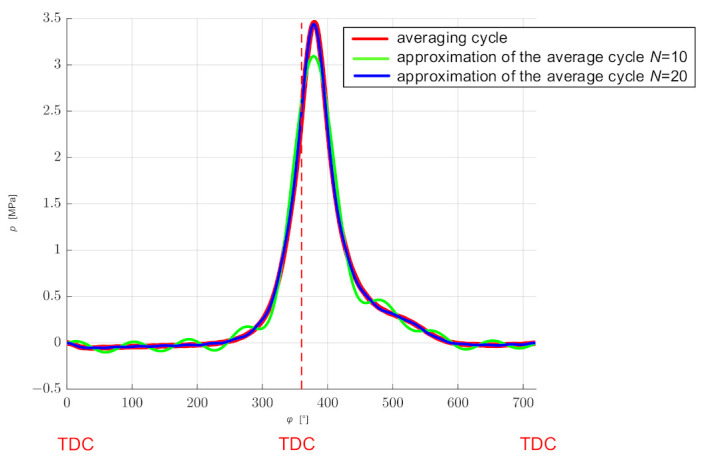
Comparison of the averaged discrete waveform with functions determined by a Fourier series with averaged coefficients for the number of components *N* = 10 and *N* = 20.

**Figure 8 sensors-21-07885-f008:**
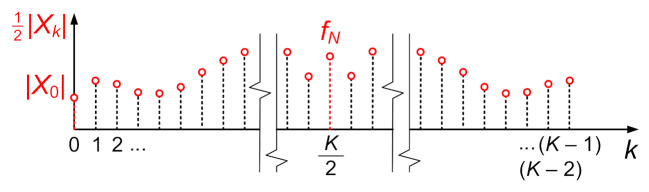
Module of an example complex spectrum calculated by means of the DFT (FFT) algorithm.

**Figure 9 sensors-21-07885-f009:**
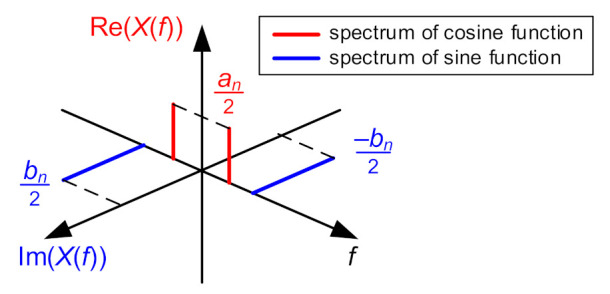
An example two-sided complex spectrum for harmonic components described by sine and cosine functions.

**Figure 10 sensors-21-07885-f010:**
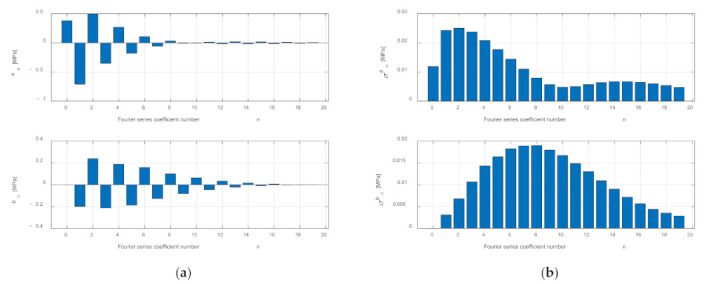
Determined coefficients *a_n_* and *b_n_* of a Fourier series after averaging (**a**) and determined values of standard deviations *σa_n_* and *σb_n_* (**b**) for 100 consecutive cycles.

**Figure 11 sensors-21-07885-f011:**
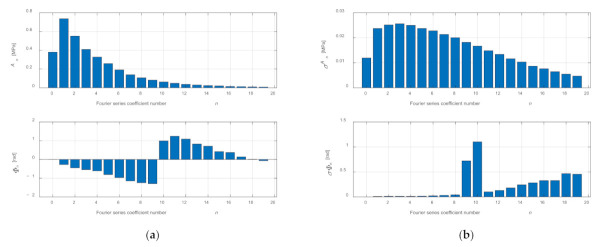
Determined coefficients *A_n_* and *Φ_n_* of a Fourier series after averaging (**a**) and determined values of standard deviations *σA_n_* and *σ**Φ_n_* (**b**) for 100 consecutive cycles.

**Figure 12 sensors-21-07885-f012:**
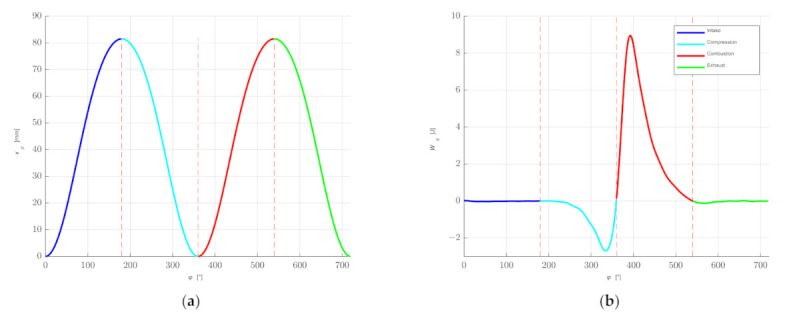
The displacement of the piston *x_p_* depending on the angle of rotation for the engine under test (**a**) and the *W_i_k_* values for subsequent angles of rotation of the shaft (**b**).

**Figure 13 sensors-21-07885-f013:**
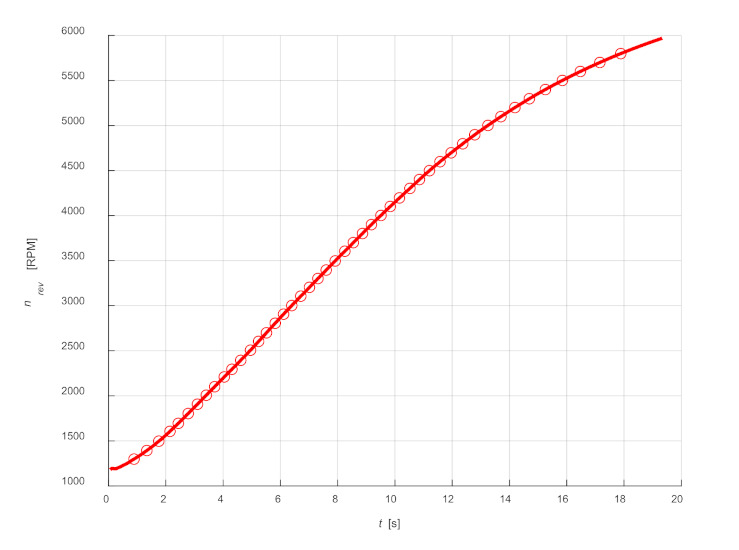
Waveform of changes in rotational speed with the values selected for further analysis.

**Figure 14 sensors-21-07885-f014:**
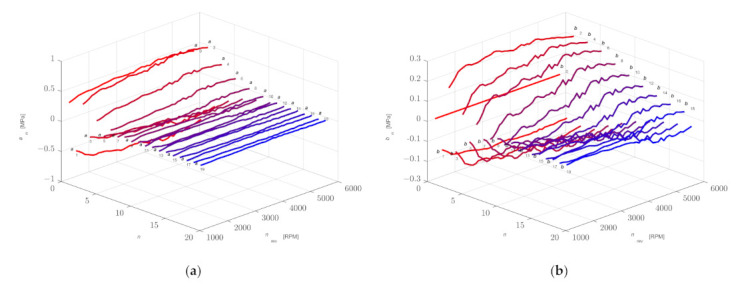
Determined values of coefficients *a_n_* (**a**) and *b_n_* (**b**) of a Fourier series as a function of rotational speed.

**Figure 15 sensors-21-07885-f015:**
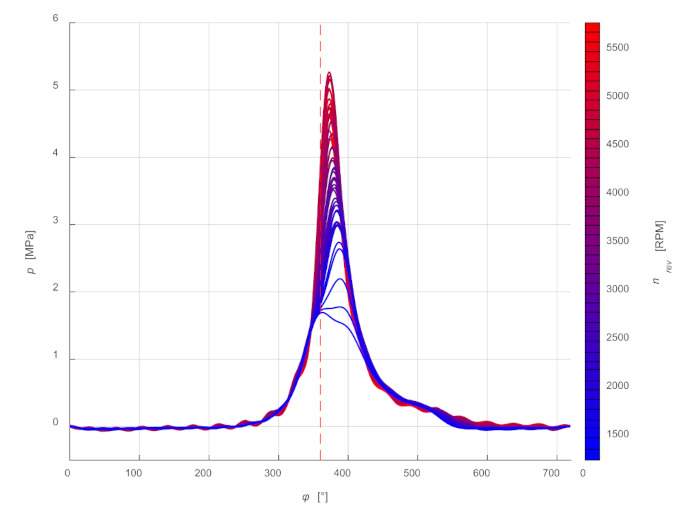
Comparison of pressure changes, determined by a Fourier series for selected rotational speeds.

**Figure 16 sensors-21-07885-f016:**
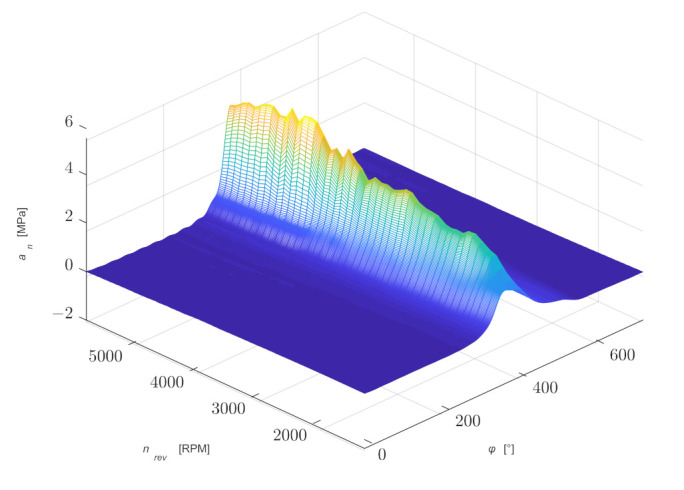
Waveforms of pressure changes as a function of the shaft rotation angle and rotational speed determined by a Fourier series.

**Figure 17 sensors-21-07885-f017:**
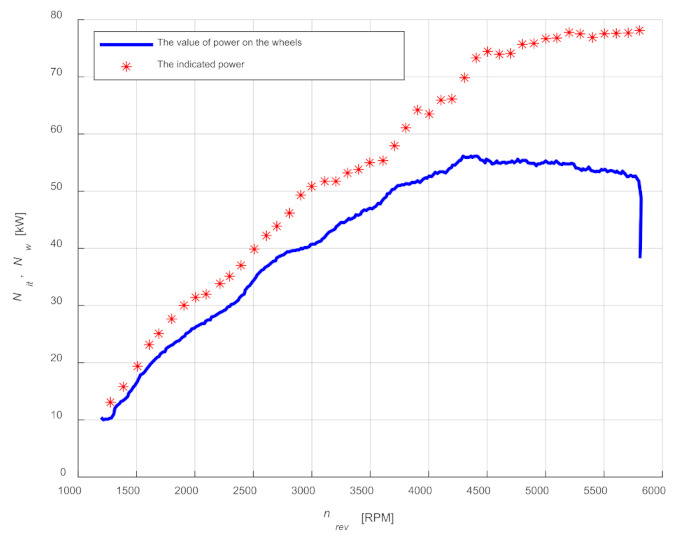
A comparison of the power values on the wheels with the determined values.

**Table 1 sensors-21-07885-t001:** Relative error values in determining time between successive pulses for the sensor of the top dead center of the piston.

Rotational Speed *n_rev_* [rpm]	Time between Successive Pulses *t_i_*–*t_i_*_−1_ [ms]	Relative Error of Time Determination [%]
1000	60	0.0127
2000	30	0.0254
3000	20	0.0381
4000	15	0.0509
5000	12	0.0636
6000	10	0.0763

**Table 2 sensors-21-07885-t002:** Values of relative errors in determining time between successive pulses for the sensor of the crankshaft position.

Rotational Speed *n_rev_* [rpm]	Time between Successive Pulses *t_j_*–*t_j_*_−1_ [μs]	Relative Error of Time Determination [%]
1000	166.67	4.578
2000	83.33	9.155
3000	55.56	13.733
4000	41.67	18.311
5000	33.33	22.888
6000	27.78	27.466

## Data Availability

Not applicable.

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
