# Peer review of "Modeling the Influence of Engine Dynamics on Its Indicator Diagram"

_sensors, 2021, doi:10.3390/s21237885_

Round 1
Reviewer 1 Report
The manuscript submited aims at evaluating torsional effects on the crankshaft of piston internal cobustion engines out of the analysis and prediction of pressure cylinder signals.
The method described and developed to model pressure changes is based on Fourier transform and statistical treatment of its coefficients.
The pressure signals recorded and analysed concern an acceleration of engine, seemingly at constant load or at idle. Authors should complete information on the torque evolution during the tests and what consequences a variation of load could influence the analysis and modeling they have carried.
Although the study is of interest, it is not evident what novelty is involved by the Fourier analysis, as developed, in comparison to number of works that have modeled and analysed cylinder pressure signals. Authors conclude (line 348) that future work will allow (only in future ?) to indentify the adopted model and globaly the major achievements of the work seem still under development.
In its present form the manuscript lacks of originality and completeness. It is not evident from it how pressure cylinder in a larger range of functionning than a simple accceleration will be correctly predicted by the method adressed.
Author Response
Thank you for your comments.
comment 1:
The pressure signals recorded and analysed concern an acceleration of engine, seemingly at constant load or at idle. Authors should complete information on the torque evolution during the tests and what consequences a variation of load could influence the analysis and modeling they have carried.
response:
The load was carried out by the chassis dynamometer control system. For constant operating conditions, the load was adjusted to maintain the set speed. When accelerating, the acceleration was adjusted based on the accelerator pedal setting.
comment 2:
Although the study is of interest, it is not evident what novelty is involved by the Fourier analysis, as developed, in comparison to number of works that have modeled and analysed cylinder pressure signals. Authors conclude (line 348) that future work will allow (only in future ?) to indentify the adopted model and globaly the major achievements of the work seem still under development.
response:
Of course, the mere application of Fourier analysis is not new. However, we did not find articles in which a description of changes in pressure waveforms as reported at changing rotational speed (acceleration) was presented. The presented modelling method is to have practical application, e.g. in the described process of simulating torsional vibrations of the engine crankshaft.
comment 3:
In its present form the manuscript lacks of originality and completeness. It is not evident from it how pressure cylinder in a larger range of functionning than a simple accceleration will be correctly predicted by the method adressed.
response:
The purpose of the presented method of describing pressure changes is to create in the model of torsional vibrations of the shaft similar engine operating conditions to tests on the chassis dynamometer: the driving force derived from the pressure acting on the piston and the load regulated in the model by the PID regulator based on changes in rotational speed. This is described in more detail in the revised article.

Reviewer 2 Report
this paper presents a proposal for a model of pressure changes in the combustion chamber of an engine as a function of rotational speed and the angle of the crankshaft position angle. The model parameters are determined on the basis of measurements carried out during acceleration on an engine test stand or chassis dynamometer. Its essence is the use of a Fourier series as a continuous function to describe pressure changes in one cycle of work. My comments are listed below:
- This article is lack of novelty, the author did not show the importance or the necessary of this methodology.
- the authors did not introduce that whether this method can be used in other application scenario: such as in diesel engines or Natural gas engines.
Author Response
Thank you for your comments.
comment 1:
This article is lack of novelty, the author did not show the importance or the necessary of this methodology.
response:
In the revised article, we described the purpose of the presented modelling description. Of course, the mere application of Fourier analysis is not new. However, we did not find articles in which a description of changes in pressure waveforms as reported at changing rotational speed (acceleration) was presented. The presented modelling method is to have practical application, e.g. in the described process of simulating torsional vibrations of the engine crankshaft.
comment 2:
the authors did not introduce that whether this method can be used in other application scenario: such as in diesel engines or Natural gas engines.
response:
Of course, It can also be used to analyse the impact of the fuel used on the acceleration dynamics. Since the tested engine is also adapted to other fuels, tests with CNG and LNG fuel have also been carried out, but this will be the topic of another article.

Reviewer 3 Report
1) The Abstract is too general and only descriptive. In the Abstract the Authors should add some of the most important results obtained in this research (its exact values), which cannot be found in other literature. Such addition will highlight the novelty of the presented paper already in the Abstract – at the moment from the abstract cannot be clearly seen novel elements which are obtained in this paper. Therefore, the Abstract requires re-arrangement and addition of the most important results obtained in the research.
2) Section 2 – In this section the Authors should present at least general specifications of the engine – at the moment any exact specifications of the analyzed engine are completely missing.
Secondly, in this section should be described and presented measurement setup, producer, type and again at least general specifications of each used measuring device.
Such general and overall presentation of the investigated engine and measuring equipment is not sufficient and acceptable (especially due to the fact that in the journal Sensors anyone will expect a detail presentation of investigated engine and especially a detail presentation of measuring equipment).
3) In the paper should be added a Nomenclature inside which will be listed and explained all abbreviations, symbols and markings used throughout the paper text. A Nomenclature will notably improve reading experience because all abbreviations, symbols and markings will be placed in one place.
4) In the titles of Figures 10, 11 and 14 should be specified what represents part (a) and (b) of each figure (as presented in Figure 12).
5) The English is clear and understandable, but it should be improved in many sentences. Also, some typing mistakes occur in the text. Please, perform a careful check of the English and perform corrections throughout the paper text.
6) As the Abstract, the Conclusions section should also be improved with the most important obtained results (its exact values). Also the Conclusions seem to be too descriptive and general, without any details obtained in the presented analysis (at least in my opinion).
7) The scientific novelty and contribution of this paper to the specific research field is not clear (at least to me). The Authors proved that the engine in-cylinder pressure change can be tracked by using Fourier series with proper coefficients. It is mentioned for what such Fourier series can be used – but there is missing any exact example, or more of them (it is proved only that the in-cylinder pressure can be tracked). There are no any results related to torsional and flexural vibrations, regulation parameters of the analyzed engine during its operation, etc. The Authors only discusses for what obtained Fourier series can be used – but any results are missing. Therefore, my opinion is that the Authors should much better and more detail explain the novelty of this paper and its importance to this research field.
Final remarks: This is an interesting paper which can be of importance in this research field, but the paper requires revision (according to above mentioned comments) and resolving all the mentioned concerns. Please, put a special attention to presenting engine specifications, details of measurement setup and measuring equipment as well as on much better and more detailed explanation of research novelty and importance to this field.
Author Response
Thank you for your comments.
comment 1:
The Abstract is too general and only descriptive. In the Abstract the Authors should add some of the most important results obtained in this research (its exact values), which can not be found in other literature. Such addition will highlight the novelty of the presented paper already in the Abstract – at the moment from the abstract cannot be clearly seen novel elements which are obtained in this paper. Therefore, the Abstract requires re-arrangement and addition of the most important results obtained in the research.
response:
We have corrected the abstract, in which we explained m.in the purpose of the presented work and the description of the presented issues. We hope that in its current form the abstract is acceptable.
comment 2:
Section 2 – In this section the Authors should present at least general specifications of the engine – at the moment any exact specifications of the analyzed engine are completely missing.
Secondly, in this section should be described and presented measurement setup, producer, type and again at least general specifications of each used measuring device.
Such general and overall presentation of the investigated engine and measuring equipment is not sufficient and acceptable(especially due to the fact that in the journal Sensors anyone will expect a detail presentation of investigated engine and especially a detail presentation of measuring equipment).
response:
We provided information on the tested engines and the chassis dynamometer used. We have completed the description of the components of the research equipment.
comment 3:
In the paper should be added a Nomenclature inside which will be listed and explained all abbreviations, symbols and markings used throughout the paper text. A Nomenclature will notably improve reading experience because all abbreviations, symbols and markings will be placed in one place.
response:
As suggested, nomenclature was added.
comment 4:
In the titles of Figures 10, 11 and 14 should be specified what represents part (a) and (b) of each figure (as presented in Figure12).
response:
Corrected descriptions under Figures 10, 11 and 14.
comment 5:
The English is clear and understandable, but it should be improved in many sentences. Also, some typing mistakes occur in the text. Please, perform a careful check of the English and perform corrections throughout the paper text.
response:
We tried to correct errors and change incorrect returns.
comment 6:
As the Abstract, the Conclusions section should also be improved with the most important obtained results (its exact values). Also the Conclusions seem to be too descriptive and general, without any details obtained in the presented analysis(at least in my opinion).
response:
The Conclusions have been reworded and supplemented. We hope that in their current form Conclusions are acceptable.
comment 7:
The scientific novelty and contribution of this paper to the specific research field is not clear (at least to me). The Authors proved that the engine in-cylinder pressure change can be tracked by using Fourier series with proper coefficients. It is mentioned for what such Fourier series can be used – but there is missing any exact example, or more of them (it is proved only that the in-cylinder pressure can be tracked). There are no any results related to torsional and flexural vibrations, regulation parameters of the analyzed engine during its operation, etc. The Authors only discusses for what obtained Fourier series can be used – but any results are missing. Therefore, my opinion is that the Authors should much better and more detail explain the novelty of this paper and its importance to this research field.
response:
In the revised article, we described the purpose of the presented modelling description. Of course, the mere application of Fourier analysis is not new. However, we did not find articles in which a description of changes in pressure waveforms as reported at changing rotational speed (acceleration) was presented. The presented modelling method is to have practical application, e.g. in the described process of simulating torsional vibrations of the engine crankshaft.

Round 2
Reviewer 1 Report
Although authors have completed the aims and purpose of the work, it is still not clear how the study proposed will or can be used in "modeling the forces forcing the movement of the piston_cranck system in simualtion" nor if it achieves substantial improvement in that direction. Fourier analysis of pressure signals is of interest among many temptatives used to extract information, but the question arises about the sense and utility of the waves studied during acceleration compared to other modeling of piston engine signals. Presenting a novel analysis might be of interest if the application and implementation of it in an aim like studying "the impact of the fuel used on the dynamics of car's acceleration" was not so speculative and the field adressed was already not so massively like combustion engines and piston pressure analysis. Also, even simple combustion models manage to catch work prediction with higher precision that what is presented in figure 17, a discussion on advantages of authors model compared to other techniques might better justify the interest of the method. But nevertheless a demonstrated implementation of the method in cranck forces or fuel consumption modeling seems necessary to distinguish the work from many other attempts to extract information from pressure signals.
Authors have greatly improved the data concerning experimental conditions and provided information on the devices used.
Author Response
comment 1:
Although authors have completed the aims and purpose of the work, it is still not clear how the study proposed will or can be used in "modeling the forces forcing the movement of the piston_cranck system in simualtion" nor if it achieves substantial improvement in that direction.
Response:
One of the many model variables in the Matlab Simulink system are: the angle of rotation of the shaft and the angular velocity determined in the subsequent integration steps. On this basis, it is possible to predict pressure changes for one cycle (two revolutions). For each cylinder, such calculations shall be carried out independently at the start of compression (start of the cycle), taking into account the random nature of the process. The pressure values thus obtained shall be converted into the axial force acting on the piston. We did not post such detailed information, believing that it could be the content of another article.
comment 2:
Fourier analysis of pressure signals is of interest among many temptatives used to extract information, but the question arises about the sense and utility of the waves studied during acceleration compared to other modeling of piston engine signals. Presenting a novel analysis might be of interest if the application and implementation of it in an aim like studying "the impact of the fuel used on the dynamics of car's acceleration" was not so speculative and the field adressed was already not so massively like combustion engines and piston pressure analysis. Also, even simple combustion models manage to catch work prediction with higher precision that what is presented in figure 17, a discussion on advantages of authors model compared to other techniques might better justify the interest of the method. But nevertheless a demonstrated implementation of the method in cranck forces or fuel consumption modeling seems necessary to distinguish the work from many other attempts to extract information from pressure signals.
Response:
We considered using other models that describe the operation of an internal combustion engine. The problem lies in the possibility of including in the model other elements of the system such as the dynamometer control system or the engine control system with their mutual couplings. Therefore, despite some limitations of the presented model, we decided to use as much information from the measurements as possible. To minimize errors, it is planned to simultaneously record the torsional vibrations of the shaft and the effective pressure in the engine, but for such measurements we must adapt the VR6 3.6 FSI engine.
We treat the study of the impact of fuel on the dynamics of car acceleration as an interesting application of the presented method, but we are aware that it should be checked in practice.
The determination of the indicated power and the comparison with the power determined by the dynamometer control system (Figure 17) serves only a qualitative comparison (similarity of waveforms). It is not a necessary element to develop the proposed model. The power characteristics determined by the dynamometer software are performed on the basis of torque and angular velocity measurements on the dynamometer rollers (power on wheels). Due to the efficiency of the drive system components and the rolling resistance, its values must be less than the indicated power. Both characteristics are fraught with errors, but of a different nature.
The main purpose of the proposed method is to use in torsional vibrations simulation studies of the crankshaft during acceleration. In this case, it was necessary to develop a model that, despite its simplicity, would be sufficiently accurate. In our opinion, these two extreme conditions have been reconciled. As we have already mentioned, we did not find a publication describing the changes in pressure waveforms that occur at increasing rotational speed (acceleration). Other possible uses should be considered as proposals to consider.
Reviewer 3 Report
The Authors have properly addressed all my comments. Required modifications/additions were performed.
The scientific novelty is now clear (at least to me). It seems that this article can be a good guideline for further research. I hope that this article will be a beginning of further research related to this topic.
My opinion is that the article is sufficiently improved and that it can be published in a presented form (revised paper version).
Author Response
Thank you for your pertinent remarks, thanks to which the reader's comprehensibility of the corrected text has increased.